# Environmental Impacts of High-Quality Brazilian Beef Production: A Comparative Life Cycle Assessment of Premium and Super-Premium Beef

**DOI:** 10.3390/ani13223578

**Published:** 2023-11-20

**Authors:** Henrique Biasotto Morais, Luis Artur Loyola Chardulo, Welder Angelo Baldassini, Isabella Cristina de Castro Lippi, Gabriela Belinassi Orsi, Clandio Favarini Ruviaro

**Affiliations:** 1UNESP–FCAV, Jaboticabal 14884-900, SP, Brazil; 2UNESP–FMVZ, Botucatu 18618-681, SP, Brazil; 3UFGD–FCA, P.O. Box 364, Dourados 79804-970, MS, Brazil

**Keywords:** beef cattle, carbon footprint, eutrophication potential, feedlot system, global warming

## Abstract

**Simple Summary:**

The environmental impacts and greenhouse gas (GHG) emissions have been evaluated across studies due to global climate change. Livestock production systems may be responsible in part for the increase in GHGs and, consequently, the environmental impacts. This study analyzed the environmental inputs and outputs and economic factors of a beef cattle feedlot system. Using a partial Life Cycle Assessment approach, we provide more information for consumers, researchers, scientists, and government officials about the environmental impacts arising from the production of premium and super-premium beef. We found that feed production is the most important source of GHGs, and diesel fuel plays a minor role in the environmental impacts. Finally, increased marbling degree in meat can lead to an increase in environmental impacts.

**Abstract:**

When individual purchasing power increases in society, there is a trend toward a quantitative and qualitative increase in the consumption of products. Considering the magnitude of beef production in Brazil, environmental impacts are important factors for the domestic and international markets. This study assessed a Brazilian feedlot system characterized by high animal welfare standards that produces high-quality beef that is more marbled than that produced in grass-fed systems. We assessed the environmental impacts and compared premium and super-premium beef produced in a feedlot system using a partial Life Cycle Assessment (LCA). Data were collected directly on the property analyzed, considering environmental inputs and outputs and economic factors associated with the production stages of each line (premium versus super-premium). The results show that high-quality beef has, beyond a greater financial cost, an environmental cost, with the super-premium line producing a 286% higher carbon footprint, 297% more eutrophication, and three times higher acidification potential and land use than the premium line. The results of the environmental impacts agree with the results of production costs, reflecting a 282.82% higher production cost in super-premium than in premium animals. Footprints of 5.0323 kg, 4.7746 kg, and 8.8858 kg CO_2_ eq./kg live weight gain at the feedlot were found in the three lines.

## 1. Introduction

Brazil has the largest commercial cattle herd in the world and is the biggest beef exporter, with 2.48 million tons exported in 2021 to over 180 countries [1]. The country is also the second largest consumer of beef behind the United States. Most of the beef produced in Brazil comes from cattle raised on pasture [2], occupying about 20% of the country’s territory [3]. In 2021, the percentage of confined animals at slaughter was 17.19%, with 6 million heads finished in feedlots [1].

Nellore animals (*Bos indicus*) account for the majority of Brazilian cattle, pasture-raised for meat production purposes [4]. This zebu breed is characterized by the production of carcasses with a low fat content and marbling—intramuscular fat (IMF). However, feedlot-finished animals produce a better-quality carcass [5], especially when crossbred with European breeds (*Bos taurus*) [6]. The use of crossbred animals (*Bos taurus* × *Bos indicus*) in drylots and early slaughtering are widely used actions to improve productiveness and meat quality [7], especially tenderness and IMF. Also, crossbred females and castrated males [8] are progressively used as consumers are willing to pay higher prices for meat products with better quality [9].

Consumers look for a high quality and variety of meat products, favoring niche markets [10,11]. Brazilian consumer behavior has been influenced by the country’s economic evolution in recent decades, heightening the population’s income and changing purchasing patterns. As individual purchasing power increases, there is a trend towards a quantitative and qualitative increase in the consumption of products [12] with a consequent increase in production to meet this demand [13,14].

Since beef is often seen as the most polluting food [15], consumers’ purchasing decisions are influenced by these impacts. Considering the magnitude of beef production in Brazil, environmental impacts are important factors for the domestic and international markets. Methane (CH_4_) is an important greenhouse gas (GHG) which contributes to climate change like carbon dioxide (CO_2_) and nitrous oxide (N_2_O). Nitrogen, on the other hand, promotes the eutrophication of freshwater. These environmental impacts can be measured directly or assessed indirectly, using methods such as the Life Cycle Assessment (LCA). This method is used and recognized worldwide to compute the impacts or potential impacts of products. In addition, the LCA allows the comparison of different production systems by classifying them according to their performance [16].

Therefore, the aim of this study was to assess the environmental impacts of premium and super-premium beef produced in a feedlot using a partial LCA. For this purpose, we assessed the environmental impact related to feeding and feed production and the enteric emissions of animals from premium and super-premium (premium (P), super-premium Angus (SPAn), super-premium Wagyu (SPW)) lines of beef in a feedlot, highlighting global warming potential, freshwater eutrophication, soil acidification, and land use.

## 2. Materials and Methods

The methodology adopted to determine the environmental impacts of gaseous emissions and to compare premium and super-premium beef produced in a feedlot system was an LCA. Primary data collected directly from the property analyzed, considering environmental inputs and outputs (emissions) related to the production stages of each line, were used. The Intergovernmental Panel on Climate Change (IPCC) Guidelines for National GHG Inventories provided the base equations used for animals and transport. The equations are presented in the Appendix A.

### 2.1. Characterization of the Production System

This research used data from the Beef Passion feedlot located in the municipality of Nhandeara, São Paulo, Brazil, located at 20°38′45.2″ south latitude and 50°01′56.8″ west longitude (Figure 1). According to the Köppen–Geiger classification, the climate is tropical savanna, summers are hot and muggy, and winters are warm and dry [17].

The property produces animals with high-quality carcasses of two beef lines: “Oba premium” (P) and “Beef Passion super premium” (SPAn and SPW). In both, animals are kept in the feedlot in order to produce beef with a high marbling score according to the AUS-MEAT marbling score, which ranges from 0 to 9 and indicates the amount of marbling in the Longissimus thoracis et lumborum muscle—between the 12th and 13th ribs [18,19]. Premium animals had a marbling score from 1 to 3, while super-premium animals had a marbling score of 4 or higher.

Premium animals are acquired from other farms and are crosses between the Nellore and Angus breeds. In this line, about 3440 animals are slaughtered annually. The predominant breed of the Beef Passion super-premium Angus animals is Aberdeen Angus (⅝ or ¾) or Wagyu (½), and the farm sends 441 and 220 animals of each line to the slaughterhouse yearly (Table 1).

In this production system, the cattle are transferred to the drylot at 17 months of age, premium animals with 300 kg and super-premium animals with 350 kg of live weight. All animals are kept in ground paddocks with troughs where all feed is provided. During this phase, the pens measure 750 m^2^ (36 paddocks, 55 animals in each, corresponding to 13.64 m^2^ per animal). The super-premium line is moved to the “bovine spa” in the last 120 days, receiving diet 3 in these pens. The difference between the pens of the bovine spa and the regular pens is the shade and the environmental enrichment with music.

The nutritional composition of the three diets is exhibited in Table 2. At the beginning of the feedlot phase, all animals eat an adaptation and growth diet (diet 1) with chopped sugar cane, dry ground corn, pelleted citrus pulp, and commercial protein concentrate. All cattle receive diet 1 for 21 days. After this period, all lines are fed diet 2, which is composed of corn germ, corn silage, pelleted citrus pulp, dry ground corn, moist citrus pulp, chopped sugar cane, peanut bran, commercial mineral core, and urea.

Afterward, the super-premium line is changed to a third diet. Meanwhile, the animals are housed in the so-called “bovine spa”, where they have constant access to music and shading in order to provide auditory enrichment. Diet 3 is composed of corn silage, corn germ, dry ground corn, moist citrus pulp, soybean grain, commercial mineral core, peanut bran, and urea. All super-premium animals receive this feed for a minimum of 120 days or until they reach the ideal slaughter condition when the live weight is generally about 730 kg for Angus and 650 kg for Wagyu animals.

### 2.2. System Boundaries

The boundary was defined from ‘the cradle to the farm gate’. In other words, all agricultural production phases. Thus, all externalities ranging from raw material extraction for production to obtaining an animal ready for slaughter were considered. The agricultural inputs and processes used for animal production were feed (grains and roughage), mineral supplements, energy, diesel oil, and fertilizers. Infrastructure and drugs were not considered due to the shortage of data. Only beef was treated as the output, without using allocation systems for co-products. Data from the EcoInvent v. 3.7 [22] and the LCA food database (Life Cycle Inventory) incorporated into the software SimaPro were used for all inputs. The data from the EcoInvent database are attributional LCA estimates.

Changes in land use were not considered and neither were emissions from animals before transfer to the feedlot nor emissions from the transport of purchased animals until they arrived at the property. The period considered was from the entering of the animals in the feedlot to their sale to the slaughterhouse.

Dry matter intake (DMI) was predicted using equation 10.17 from the Guidelines for National GHG Inventories [23]. Predictions were made for each line and all phases in the drylot; the values ranged between 2.03 and 2.27% of body weight.

Emissions from feed production were multiplied by the total feed intake of the animal, divided by the weight gained by the animal in the feedlot.

### 2.3. Functional Unit

The functional unit selected in this study was “one kilogram of live weight gained in the feedlot” in order to compare the environmental performance of this feedlot operation with other systems. The results are additionally reported as kilograms of carcass gained in the feedlot. Carcass weights were measured by the slaughterhouse, in accordance with normal federal inspection procedures for the meat industry, and the data were provided to the authors by the animals’ owner.

Feedlot emissions, enteric emissions of the animals, and the respective inputs used on the farm (production of diet ingredients, diesel fuel used for the transportation of diet ingredients) were assessed as carbon dioxide equivalent.

The emissions from the production of the feeds were calculated with SimaPro (v.9.2), through LCA data on all the ingredients of the feeds (EcoInvent v.3.7), using Brazilian studies to be more reliable with regard to the impacts, since the ingredients used in the feedlot are all produced in Brazil. Emissions from feed production were multiplied by the total feed intake of the animal, divided by the weight gained in the drylot.

### 2.4. Categories and Impact Assessment

The global warming category was elected during the phase of impact assessment and consisted of all GHGs emitted during the production of the product converted to CO_2_ equivalent in relation to the selected functional unit. GHG emissions from the cattle were estimated according to the Tier 2 approach of Chapter 10, volume 4 of the IPCC [23]. The Tier 2 method uses data on the animal, animal productivity, diet digestibility and energy, and system details to produce a more detailed estimate of intake, predicting methane generated from enteric fermentation and CH_4_ and N_2_O emissions from dung [23]. According to the IPCC, enteric methane emissions can have an uncertainty of ±20% on the emission factor, and for the manure management emission factors, it is ±30%.

The 100-year time horizon GWP relative to CO_2_ equivalence used was 28 for methane and 265 for nitrous oxide, according to [24]. Other environmental impacts, like freshwater eutrophication potential, terrestrial acidification potential, and land use were analyzed and estimated using data from the EcoInvent v. 3.7 and the LCA food database (Life Cycle Inventory) incorporated into SimaPro LCA software (v. 9.2). Regarding the assessment of each line, the evaluation period consisted of the time required for an animal to reach slaughter weight, from entry into confinement until the last day of feeding.

Three different products were compared, the differences between them being the diets (number of days each diet was ingested, DMI of each diet, gross energy of each diet) and animal weights (weight at the beginning of confinement, final weight at slaughter, mature body weight, carcass weight, carcass yield). Equations from Chapter 10, volume 4 of the IPCC [23] (10.3, 10.6, 10.14, 10.15, 10.16, 10.17, 10.21) were used to assess enteric methane emissions from animals. Equations 10.22, 10.23, 10.24, 10.25, and 10.30 were used to estimate N_2_O production from manure management. All of them are provided in the Appendix A.

### 2.5. Transportation Emission

The transport emissions were calculated solely for feed deliveries made to the farm. The weight of the load determined the size of the truck, which directly impacted fuel efficiency. Fuel consumption data for each truck type were obtained from [25]. The transportation emissions considered both the transport of the loaded truck and its return trip when empty to the original point of departure. Delivery distances ranged from 3 km (corn silage produced on the farm) to 435 km. The emission factor used was 2603 kg CO_2_ eq. for each liter of diesel fuel used [26].

### 2.6. Economic Analysis

The software RLM Corte v.3.3 (https://www.rlm.app.br/principal, accessed on 1 October 2023) was adopted to evaluate all diets using the Tropicalized NRC system [21], including total daily cost (USD/animal/day), weight gain cost (USD/kg), price of animal (USD/animal), feedlot cost (USD), and diet price (USD/ton). The original matter costs (USD/ton) of the ingredients were corrected according to quotations obtained in São Paulo state in 2023.

## 3. Results

The results of environmental impacts are presented in Table 3. Isolated carbon footprints from diet production and from enteric fermentation are shown.

The median daily feed consumption of the cattle was 2.3, 2.1, and 2% of the live weight for premium, super-premium Angus, and super-premium Wagyu animals (Table 4).

Enteric fermentation (CO_2_ eq./LWG) of SPAn was 14.01% above that of P animals, and that of SPW was 77.78% superior to that of P animals. The carbon footprints were 5.001 kg CO_2_ eq./LWG for premium, 4.743 kg CO_2_ eq./LWG for super-premium Angus, and 8.8325 kg CO_2_ eq./LWG for super-premium Wagyu lines during finishing. The terrestrial acidification was 24.71, 28.22, and 47.30 g SO_2_ eq./kg live weight gained in the feedlot for premium (100%), SPAn (114.19%), and SPW (191.37%) animals. The freshwater eutrophication assessed for each beef line was 1.25, 1.38, and 2.30 g PO_4_ eq./kg live weight gained in the feedlot, as 100% for P, 110.11% for SPAn, and 183.34% for SPW animals.

The diet of the premium line produced 4.038 kg CO_2_ eq./kg LWG in the feedlot, the super-premium Angus diet emitted 3.644 kg CO_2_ eq./kg LWG in the feedlot, and the super-premium Wagyu diet emitted 6.471 kg CO_2_ eq./kg LWG in the feedlot. These emissions, reported as kg CO_2_ eq./kg carcass weight gained in the feedlot, were 7.604, 6.662, and 11.682, respectively.

Each kg of transported ingredient emits on average 6 g of CO_2_ eq. The total amount of diesel fuel used was 28,738 L, estimated based on the consumption per ton transported by heavy, semi-heavy, and light trucks.

Economic analysis (Table 5) and the environmental impacts were used to compare the premium versus the SPAn and SPW lines. The cost (USD/LWG) of the P line was considered to be 100%, while those of the SPAn and SPW lines were 134.31% and 196.58%, respectively. The GHG emission from enteric fermentation (CO_2_ eq./LWG) of P animals was considered to be 100%; SPAn was 114.01% and SPW was 177.78%. The terrestrial acidification (SO_2_ eq./LWG) for P, SPAn, and SPW animals was 100%, 114.19%, and 191.37%, whereas the freshwater eutrophication category (PO4 eq./LWG) was found to be 100%, 110.11%, and 183.34%, respectively.

The production cost (USD/@) of the animals in the P, SPAn, and SPW lines was USD 91.43, USD 133.23, and USD 200.00, corresponding to 100%, 145.72%, and 218.75%, respectively (one @ is equal to 15 kg). When evaluating the cost per kg gained in the feedlot, we found USD 127.28 for P animals (100%), USD 140 for SPAn animals (109.99%), and USD 360.11 for SPW animals (282.92%).

## 4. Discussion

The utilization and correlation of LCA studies may be limited by disparity in boundaries, objectives, or functional units [16]. Generating LCA models that consider distinct management strategies and technologies is critical because of the increasing consumer interest in sustainable beef production, as well as the need for a complete analysis of these different systems [11].

Niche markets are composed of distinct products that serve a specific clientele that is willing to pay more for the product [27]. All the beef produced on the farm that is the object of this study is sold in upscale restaurants and in the specialized stores of the producers themselves, which justifies the high costs of feeding the animals and the high price per LWG. Table 5 describes the production cost of each line.

Beef products can be distinguished based on intrinsic or extrinsic characteristics related to meat quality or the production process [28]. Some of the intrinsic factors that can differentiate beef are tenderness, juiciness, flavor, pH, nutritional factors, color, fat cover, and marbling. The first three are the main influencers of eating quality [29], while the last three are perceived by the consumer’s eye at the time of purchase.

The extrinsic factors need certifications or label information to be perceived by consumers, like animal welfare and environmental impacts [30]. The beef in this study is visually differentiated from common beef because of the high marbling and certification that it has. Other characteristics, like animal welfare and juiciness, are highlighted by the brand but not certified on the packaging.

There has been a change in the consumers’ purchase behavior of beef that goes beyond intrinsic factors affecting eating quality, including environmental issues [15]. In cattle farming, most attention is given to the emissions of GHGs.

In contrast to the literature [11,31], our study revealed that the primary source of GHG emissions in beef production did not originate from the enteric fermentation of animals. Instead, the production of feed used for livestock feed was found to contribute the most, in line with the findings of Cole et al. [32]. This discrepancy may be attributed to the specific diet employed in the feedlot system, which predominantly relies on crop products cultivated using nitrogen fertilizers, a significant source of nitrogen emissions [33].

Environmental impacts can vary widely across the diversified production scenarios found in Brazil, especially in terms of carbon footprint and greenhouse gas emissions [34,35]. Some researchers obtained carbon footprints between 18.3 and 42.6 kg of CO_2_ eq. for each 1 kg of live weight gain [36] (in the complete beef cattle system), whereas others reached carbon footprints between 9.16 and 22.5 kg CO_2_ eq./kg of live weight gained [31] (all components of the beef production). In this study, the values were 5.0 (premium), 4.74 (super-premium Angus), and 8.18 (super-premium Wagyu) kg CO_2_ eq. per kg of live weight gained in the feedlot.

Although the results of this study are notably lower compared to the those of the aforementioned studies, it is crucial to emphasize that the beef production examined in this research is characterized by differentiation, and the focus was exclusively on the feedlot finishing stage of the animals. It should be noted that the cow–calf phase is widely recognized as the primary contributor to environmental impacts, comprising approximately 69% to 84% of the overall production cycle [37,38].

When correlating our study with others that specifically assessed the finishing period, such as [39], who reported a footprint of 6.9 kg CO_2_ eq./kg of live weight, some similitude is visible. However, it is important to note a distinct functional unit, indicating that our study has a smaller footprint when considering the live weight gained (LWG). Furthermore, when paralleling with studies that utilized the same functional unit (live weight gained in confinement), [40] reported 4.84 kg CO_2_ eq./kg of LWG, [41] obtained 7.61 kg CO_2_ eq./kg of LWG, and [42] found results of 10.16 kg CO_2_ eq./kg of LWG. These results are similar to the values obtained in our work, considering the similar boundaries and functional units employed.

The land exploited in feedlot production is smaller than that used for pasture-based production. In this study, the beef production was 3.7, 9.83, and 16.55 kg of LWG/ha/day for super-premium Wagyu, premium animals, and super-premium Angus animals, counting the land destined for crop ingredients for the animals’ diets. These results are similar to the land use measured by Vale et al. [5], from 3.23 to 23.01 kg/ha/day.

Considering only the feed production land, this area comprised 0.08 ha, 0.187 ha, and 0.282 ha for P animals, SPAn animals, and SPW animals. Productions of 0.2 to 0.8 kg of LWG/ha/day for extensive systems and 1.02 to 4.22 kg of LWG/ha/day for intensive systems were reported before [5]. Thus, feedlot systems are able to provide up to 80 times more beef per area by day than the extensive system. However, pasture-based systems may yield 13% more regular beef than an intensive system may produce super-premium beef, emphasizing that beef with a higher degree of fatness (and added value) has a high environmental cost. Such results occur due to the disparity in energy composition between fat and muscle tissues. The deposition of fat cover and marbling requires more energy intake [43], which commonly increases the days on feed.

The emissions related to total diets were 703.75 kg CO_2_ eq. for P animals, 1355.65 kg CO_2_ eq. for SPAn animals, and 2168 kg CO_2_ eq. for SPW animals. The premium line’s results were alike those reported by Stackhouse-Lawson et al. [37] (697–729 kg) but greater than those of Cole et al. [32] (620 kg). The values found to produce the diets consumed by super-premium animals (Angus and Wagyu) can be associated with their superior dry matter intake (119.5% and 229% higher than those of premium animals, respectively).

Werth [44] obtained values of 0.63, 0.58, and 0.52 kg of CO_2_ e./kg of diet, related to this work where we reached 0.393, 0.856, and 0.626 kg of CO_2_ e./kg of diet for diet 1, diet 2, and diet 3; the results are similar, although different ingredients produce different impacts on the production.

In accordance with the literature [39,41,42,45], this study also suggests that premium and super-premium beef produced in feedlots could generate fewer environmental impacts than pasture-based production, and the product has higher added value and quality, concerning animal welfare standards, using a smaller area, and employing resources effectively. Furthermore, the feedlot system enables the production of beef with marbling degrees that cannot be produced in extensive or intensive pasture systems, supplying specific market niches.

The results of three environmental impacts (enteric fermentation, terrestrial acidification, and freshwater eutrophication) agree with the results of production costs. P animals had the lowest production cost per head, as well as the lowest cost per LWG and cost per kg. The highest production cost was found in SPW animals, being 282.82% higher than that of P animals. These results were due to several factors, such as time on diet 2 (diet with the highest carbon footprint per kg), days on feed, daily weight gain, and feed conversion, which was lower in SPW animals.

Some studies describe environmental impact and economic performance as inversely correlated. Bonnin et al. [46] reported a correlation coefficient of −0.7 to support this finding. Similarly, Pedolin et al. [47] reported inverse correlations between economic and environmental impacts in livestock production. These authors also observed the presence of fixed impacts that are independent of productivity, like transportation, electricity, and the cow–calf phase. Increasing the outputs of these fixed impacts can enhance environmental performance through improved efficiency. This phenomenon is evident in our study in the SPAn line, which costs at least 2.5 times more (USD/animal) than the P line to produce both superior carcasses and meat quality. Furthermore, we suggest a drop in productivity and reduced efficiency as there are lower outputs to distribute fixed environmental costs, as observed for SPW animals compared to SPAn animals.

## 5. Conclusions

Super-premium beef (super-premium Wagyu line) showed divergent results for all impacts assessed (terrestrial acidification potential, freshwater eutrophication potential, global warming potential) and used more land for production. Premium-line animals (Oba) produced the lowest environmental impacts in terms of freshwater eutrophication potential, terrestrial acidification potential, and smallest land use.

The super-premium Angus line exhibited the smallest global warming potential due to its higher slaughter weight and median performance in all other categories. The production of animal diets made the most significant contribution to the global warming potential, enteric and manure emissions, and lastly transportation.

The findings of this study emphasize the major participation of food production in the environmental impacts of feedlot beef cattle. Furthermore, the production of premium-quality beef comes with greater production costs.

## Figures and Tables

**Figure 1 animals-13-03578-f001:**
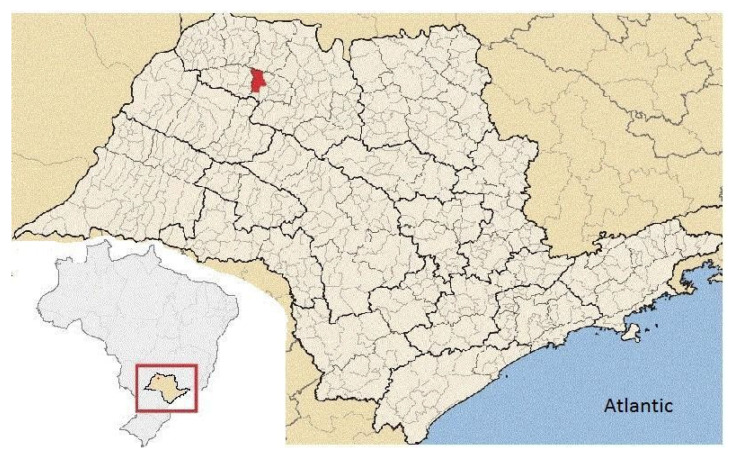
Location of the municipality of Nhandeara in São Paulo state and location of the São Paulo state in Brazil. Adapted from https://pt.wikipedia.org/wiki/Nhandeara, accessed on 1 October 2023.

**Table 1 animals-13-03578-t001:** Characterization of the production system showing the breed composition of each line, weights, duration of each phase, and pen type.

Line	Premium	SPAn	SPW
Breed composition	N × A	NA × A	NA × W
Initial weight (kg)	300	350	350
Slaughter weight (kg)	485	730	650
LWG (kg)	185	380	200
Days in diet 1	21	21	21
Days in diet 2	100	80	180
Days in diet 3	-	120	120
Pen type	Regular	Regular/Spa	Regular/Spa
Days in feedlot	121	221	321

SPAn: super-premium Angus, SPW: super-premium Wagyu, N: Nellore; A: Angus; NA: F1 Nellore × Angus; W: Wagyu; LWG: live weight gain.

**Table 2 animals-13-03578-t002:** Nutritional information and chemical composition of diets.

Description	Unit	Diet 1	Diet 2	Diet 3
Dry matter	(%)	39.46	52.88	53.85
Total digestible nutrients	(% DM)	60	65.6	73.5
Gross energy	(MJ/kg DM)	17.04	10.34	11.72
Ether extract	(% DM)	1.6	3.78	5.20
Starch	(% DM)	7.31	16.05	19.15
Crude protein	(% DM)	6.75	13.07	13.02
RDP	(% DM)	2.88	6.80	5.56
NPN	(% RDP)	57.38	69.25	66.29
NDF	(% DM)	45.47	30.39	31.80
ADF	(% DM)	27.23	5.02	3.84
Carbon footprint	kg CO_2_ eq./kg DM	0.393	0.856	0.626
Ingredients		% (DM)	% (DM)	% (DM)
Chopped sugar cane		74.12	7.00	0
Dry ground corn		15.22	11.15	16.82
Pelleted citrus pulp		6.56	13.10	0
Commercial protein concentrate		4.10	0	0
Corn germ		0	28.90	29.90
Corn silage		0	21.75	33.57
Moist citrus pulp		0	9.00	9.00
Peanut bran		0	5.34	2.50
Commercial mineral core		0	2.65	2.65
Urea		0	1.11	0.76
Soybean grain		0	0	4.80

DM: dry matter; RDP: rumen-degradable protein; NPN: nonprotein nitrogen; NDF: neutral detergent fiber; ADF: acid detergent fiber. From the formulations, composition information was obtained in CQBAL [20] and RLM [21].

**Table 3 animals-13-03578-t003:** Emissions from diet and enteric fermentation (kg CO_2_ eq.), terrestrial acidification (g SO_2_ eq.), freshwater eutrophication (g PO_4_ eq.), and carbon footprint (kg CO_2_ eq.), assessed per kg of live weight gain (LWG) in the feedlot.

Emissions	
From enteric fermentation	kg CO_2_ eq./kg LWG
Premium	0.9634
Super-premium Angus	1.0984
Super-premium Wagyu	1.7128
Total potential impacts	
Terrestrial acidification	g SO_2_ eq./kg LWG
Premium	24.7145
Super-premium Angus	28.2229
Super-premium Wagyu	47.2982
Freshwater eutrophication	g PO_4_ eq./kg LWG
Premium	1.2541
Super-premium Angus	1.3809
Super-premium Wagyu	2.2993
Carbon footprint	kg CO_2_ eq./kg LWG
Premium	5.0323
Super-premium Angus	4.7746
Super-premium Wagyu	8.8858
Land use	m^2^/kg LWG
Premium	4.34
Super-premium Angus	4.93
Super-premium Wagyu	9.40

LWG: live weight gain.

**Table 4 animals-13-03578-t004:** Zootechnical and productive indices of the three cattle lines.

Description	Premium Line	Super-Premium Line	
	Oba	Angus	Wagyu
Land use (m^2^/head)	802.4	1873.5	2819.1
Feed conversion	5.17	5.27	8.8
Total DMI (kg/head)	901.46	1961.25	2949.75
Carcass yield (% live weight)	53.1	54.7	55.4
Slaughter by year (animals)	3440	441	220

DMI: dry matter intake.

**Table 5 animals-13-03578-t005:** Production costs (total and feed) of feedlot diets.

	Diet/Line							
	Diet 1/P	Diet 1/SPAn	Diet 1/SPW	Diet 2/P	Diet 2/SPAn	Diet 2/SPW	Diet 3/SPAn	Diet 3/SPW
Gain cost (USD/@)	282.68	180.21	190.13	212.70	179.07	225.68	304.32	235.13
Daily cost (USD/animal/day)	7.96	8.57	8.57	10.24	10.22	10.74	15.96	14.03
Cost of diet (USD/animal)	232.89	245.70	245.70	1337.00	1068.00	2496.60	2233.20	2059.20
Feed cost (as fed, USD/t)	396.92	396.92	396.92	564.58	564.58	564.58	678.86	678.86
Feed cost (DM, USD/t)	1006.72	1006.72	1006.72	1067.13	1067.13	1067.13	1279.20	1279.20
	Feed cost of animal production							
		USD/animal	USD/LWG					
Premium		1191.16	6.44					
Super-premium Angus		3288.37	8.65					
Super-premium Wagyu		3796.77	12.66					

@: 15 kg; P: premium; SPAn: super-premium Angus; SPW: super-premium Wagyu; DM: dry matter; t: ton; LWG: live weight gain. Diet 1, diet 2, and diet 3 have 26:74, 62:38, and 57:43 concentrate–roughage ratios, respectively.

## Data Availability

Data available on request due to restrictions eg privacy or ethical. The data presented in this study are available on request from the corresponding author. The data are not publicly available due to the copyright of private software.

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
