# Peer review of "Environmental Impacts of High-Quality Brazilian Beef Production: A Comparative Life Cycle Assessment of Premium and Super-Premium Beef"

_animals, 2023, doi:10.3390/ani13223578_

Round 1

Reviewer 1 Report

The manuscript “Environmental Impacts of High-Quality Brazilian Beef Production: A Comparative Life Cycle Assessment of Premium and Super-Premium Beef” deals with the assessment of the environmental impact associated with different beef production lines, focused on fattening part of the beef cycle.

The topic is interesting, as the need to assess products increasingly demanded by consumers is a priority within the effort to limit the negative impacts of the food provision systems. Although this, the manuscript has different flaws that must be fixed.

The most evident one is the need to make materials and methods clearer and more complete. In particular, equations and emission factors used to compute the impact have to be reported in the manuscript (main text or supplementary materials) – reporting only the references is not sufficient. Furthermore, the ingredient composition (% DM on 1 kg dry matter) and dry matter intakes (kg DM/head/day in the different phases and as average of the whole fattening period) have to be clearly reported. Besides, what about the input-output animal flows of nitrogen and phosphorous? How did the authors compute them?

From the LCA point of view, considering only the fattening period of the beef production cycle made the study not a complete LCA but only “partial LCA”. Please fixed. Moreover, Life cycle impact assessment for acidification and eutrophication needs to be stated (CML; Recipe;….?). Besides, I do not understand how authors considered emissions from manure management and agronomical phase of feed productions (no info are reported in terms of description, inventories, emissions).

Another LCA-related issue, especially when single sample per treatment has been assessed, is the need to perform a uncertainty analysis on the possible variation in the emission factors’ values. Please provide (IPCC documentation reports the % of uncertainty associated with the different emission factors for animal and manure emissions, for example).

Moreover, I suggest to re-check the sentences formulation: e.g., in sub-chapters 2.4 and Discussion, various parts are difficult to read, and parts related to the same thing have been positioned far from each other in the text.

The authors also introduced an economic analysis of the production, but they did not discuss in complete way and, moreover, I suggest studying better the interactions between environmental and economic assessments done (and simplify table 5 since it is difficult to follow)

Specific comments:

Lines 64-65. Specify CO2 and N2O as done for methane (so carbon dioxide and nitrous oxide)

Line 67. Authors did not measure emissions. Replace with “assessed”. Check throughout the manuscript.

Lines 72-77. “premium” and “super premium” has to be introduced earlier in the Introduction.

Table 2. Check “crude energy” (I guess that it is “gross energy”) and in particular check “crude protein” (in diet 2 and diet 3 they are extremely high, too much high)

Line 160. Carcass data derived from?? Indicate if it was a data collected by authors

Chapter 2.3. why not try a functional unit based on nutritional profile, such as 1 kg protein plus fat? As the lines are set to provide beef of different quality also in terms of nutrients, the environmental comparison between lines can be biased by the dis-uniformity of the functional unit with respect to the actual product obtained from the different lines.

Lines 170-186. As author used also the land occupied per 1 kg LWG, “land occupation” needs to be reported an impact category assessed (together with global warming, acidification and eutrophication)

Lines 184-186. Seems a repetition

Table 3. Why data are disaggregated for global warming and not for other categories?

Lines 243-258. Non results reporting with respect to other categories than GWP?

Table 5. what is it “@”??? And “diary”???? Please reformulate the table to report in clear way the data about economic cost

Lines 261-262. The meaning is understandable, but the sentence can be simplified.

Lines 272-285. Can be reduced, as the manuscript did not deal with beef quality

Lines 292-298. Specify always if digits referred only to the feedlot or not

Lines 314-328. Kg/ha/day referred to what? Please also check for discourse consistency, because the different sentences seem not always logically linked

Lines 329-335. Absolute kg CO2-eq, if it is not the focus of the assessment, does not have much sense to be discussed.

Lines 346-352. Need a more in-deep discussion

Conclusions. They seem to be a bit weak.

Author Response

The manuscript “Environmental Impacts of High-Quality Brazilian Beef Production: A Comparative Life Cycle Assessment of Premium and Super-Premium Beef” deals with the assessment of the environmental impact associated with different beef production lines, focused on fattening part of the beef cycle.

The topic is interesting, as the need to assess products increasingly demanded by consumers is a priority within the effort to limit the negative impacts of the food provision systems. Although this, the manuscript has different flaws that must be fixed.

The most evident one is the need to make materials and methods clearer and more complete. In particular, equations and emission factors used to compute the impact have to be reported in the manuscript (main text or supplementary materials) – reporting only the references is not sufficient. Furthermore, the ingredient composition (% DM on 1 kg dry matter) and dry matter intakes (kg DM/head/day in the different phases and as average of the whole fattening period) have to be clearly reported. Besides, what about the input-output animal flows of nitrogen and phosphorous? How did the authors compute them?

We sincerely appreciate the valuable comments and suggestions from Reviwer.

We used tier 2 methodology for computing NO2. 
DMI was predicted for each animal group, in each diet. We reported 8 means of DMI and these values were around 2.03% and 2.27% of BW. We corrected this information in the revised manuscript.

From the LCA point of view, considering only the fattening period of the beef production cycle made the study not a complete LCA but only “partial LCA”. Please fixed. Moreover, Life cycle impact assessment for acidification and eutrophication needs to be stated (CML; Recipe;….?). Besides, I do not understand how authors considered emissions from manure management and agronomical phase of feed productions (no info are reported in terms of description, inventories, emissions).

We corrected as recommeded to "partial LCA". For acidification and eutrophication variables, were used the Recipe method in SimaPro, the same software which generates the emissions from all feed ingredients by the EcoInvent v. 3.7 inventory. For example:
Diet 1: 
Inputs from technosphere: materials/fuels
Citrus ulp dried, from drying, at plant/US Mass; Maize, at farm/BR Mass; Sugarcane [BR]|production|Alloc Def, S; Soybean meal, from crushing (solvent), at plant/BR Mass. And the respective quantity of each ingredient.

Another LCA-related issue, especially when single sample per treatment has been assessed, is the need to perform a uncertainty analysis on the possible variation in the emission factors’ values. Please provide (IPCC documentation reports the % of uncertainty associated with the different emission factors for animal and manure emissions, for example).

Corrected. We added the Uncertainty values for the EF used, thanks.

Moreover, I suggest to re-check the sentences formulation: e.g., in sub-chapters 2.4 and Discussion, various parts are difficult to read, and parts related to the same thing have been positioned far from each other in the text.

Thanks. We double-checked these sentences in the revised manuscript. 

The authors also introduced an economic analysis of the production, but they did not discuss in complete way and, moreover, I suggest studying better the interactions between environmental and economic assessments done (and simplify table 5 since it is difficult to follow)

We improved this topic in the Discussion section and added two new studies. 
Table 5 was simplified and improved as recommended. Thank you

Specific comments:

Lines 64-65. Specify CO2 and N2O as done for methane (so carbon dioxide and nitrous oxide)
Done! Thank you.

Line 67. Authors did not measure emissions. Replace with “assessed”. Check throughout the manuscript. 
Correct. Thank you.

Lines 72-77. “premium” and “super premium” has to be introduced earlier in the Introduction. 
Right, made. Thank you.

Table 2. Check “crude energy” (I guess that it is “gross energy”) and in particular check “crude protein” (in diet 2 and diet 3 they are extremely high, too much high) 
Correct, table 2 fixed, please see revised manuscript. Thank you.

Line 160. Carcass data derived from?? Indicate if it was a data collected by authors
Corrected in manuscript, thank you. Please see revised version. 

Chapter 2.3. why not try a functional unit based on nutritional profile, such as 1 kg protein plus fat? As the lines are set to provide beef of different quality also in terms of nutrients, the environmental comparison between lines can be biased by the dis-uniformity of the functional unit with respect to the actual product obtained from the different lines.
The reviewer is right. The products from the different lines are not uniform or exactly the same. In the current study, the super-premium beef line had more fat than premium. To provide this kind of standardization (e.g. functional unit in one kg of protein plus fat) and improve the comparison within the nutritional aspect, we will need additional data. However, the property does not have this information obtained experimentally with a reasonable number of repetitions, and to obtain such information, other studies would have to be carried out, including laboratory assays. 

Lines 170-186. As author used also the land occupied per 1 kg LWG, “land occupation” needs to be reported an impact category assessed (together with global warming, acidification and eutrophication)
Correct. Thank you.

Lines 184-186. Seems a repetition
Removed, thank you

Table 3. Why data are disaggregated for global warming and not for other categories?
Because we would like to emphasize the importance of diets and the animals' phenotypes in beef production, this way we could make more visible to readers where the differences in carbon footprint came from.

Lines 243-258. Non results reporting with respect to other categories than GWP? We improved these sections of results, please check the corrected version. Thank you.

Table 5. what is it “@”??? And “diary”???? Please reformulate the table to report in clear way the data about economic cost
@ is a weight Unity, equivalent to 15 kg. It’s widely used in the cattle trades. We provided legend to become clearer. Diary is like daily. We reformulated and simplified Table 5, thank you.

Lines 261-262. The meaning is understandable, but the sentence can be simplified.
Done, thank you.

Lines 272-285. Can be reduced, as the manuscript did not deal with beef quality
We made it more concise, thank you.

Lines 292-298. Specify always if digits referred only to the feedlot or not
Corrected, thank you.

Lines 314-328. Kg/ha/day referred to what? Please also check for discourse consistency, because the different sentences seem not always logically linked
Thank you, we improved this paragraph, please see revised version. 

Lines 329-335. Absolute kg CO2-eq, if it is not the focus of the assessment, does not have much sense to be discussed.
We think that the absolute emission of diets is important to emphasize the importance of improving diet's footprint when aiming to reduce environmental impacts. Thank you.

Lines 346-352. Need a more in-deep discussion 
We expanded the end of discussion. Thank you

Conclusions. They seem to be a bit weak.
We improved the Conclusion section. We sincerely appreciate the valuable comments and suggestions from Reviewer.

Reviewer 2 Report

This research assesses high-quality beef cattle feedlot systems' environmental impact and economic factors. However, multiple concerns with the methodology and results presentation led to the limited discussion section. Thus, all the concerns could be addressed with a revision. I have specific comments and suggestions as follows.

L84 specifies IPPC reference use.

L92 What is Oba? defined when used the first time.

L95 change “et” to “at”

L131 Table 2

-provides Table Feed ingredients and feed formulation of each diet.

            -Crude energy 1) Correct this is a "gross energy content" not "crude energy content" or "metabolizable energy (ME) content"? 2)if it is ME content, Diet 1 is the wrong data; Diet 2, and 3 are acceptable; pls correct.

            -Crude protein Re-check %CP estimation of diets is an outage (32.55, 27.66% for diets) of the common range (CP 12-18%), and diet 1 is too low according to beef cattle requirement NRC or Brazil recommendation.

            -RDP re-check RDP estimation because the data is out of the commendation eg. NRC or Brazilian Tables of Food Composition for Ruminants

            -NPN re-check NPN estimation because the data is an outage the commendation eg. NRC or Brazilian Tables of Food Composition for Ruminants

            -ADF re-check ADF estimation because the data is an outage the commendation eg. NRC or Brazilian Tables of Food Composition for Ruminants

L170 adds the assessment impact from purchased feed categories because feeding contributes a large portion.

L182-186 describe how you consider CO2 assessed from manure, purchased feed, fertilizer, electricity, and fuels.

L220 Table 3

1)Suggest separating result Table eg. Carbon footprint, acidification, eutrophication with differing sources impact category

2)Carbon footprint presents data that categorized GHG sources (e.g. enteric methane fermentation, manure, purchased feed, fertilizer, electricity, fuels

3) Suggest to the discussion that the values are relatively low compared with other

L235-239 provide the result in the Table 2

L2243 Table 5 cannot be read, pls correct the presentation.

L245 -249 move to GHG emission result section

L250-253 Table 5 cannot be read, pls correct the presentation.

L297-298 re-check Table 3 CFP estimation to include all main category source impact e.g.  (e.g. enteric methane fermentation, manure, purchased feed, fertilizer, electricity, and fuels.

L346 enteric fermentation change to GHG emission

L361-363 data do not show, pls present Table data.

Author Response

This research assesses high-quality beef cattle feedlot systems' environmental impact and economic factors. However, multiple concerns with the methodology and results presentation led to the limited discussion section. Thus, all the concerns could be addressed with a revision. I have specific comments and suggestions as follows.

L84 specifies IPPC reference use.

We sincerely appreciate the valuable comments and suggestions from the Reviewer. We now provide a Supplementary Table. Please, see the revised manuscript. 

L92 What is Oba? defined when used the first time.
Corrected. OBA is one of the two beef lines in our study. Please, see revide manuscript (L93-94).

L95 change “et” to “at”
We kept the "et" because refers to scientic name of this muscle as desbrided in Arroyo, C., Lascorz, D., O'Dowd, L., Noci, F., Arimi, J., & Lyng, J. G. (2015). Effect of pulsed electric field treatments at various stages during conditioning on quality attributes of beef longissimus thoracis et lumborum muscle. Meat Science99, 52-59.

L131 Table 2

-provides Table Feed ingredients and feed formulation of each diet.

            -Crude energy 1) Correct this is a "gross energy content" not "crude energy content" or "metabolizable energy (ME) content"? 2)if it is ME content, Diet 1 is the wrong data; Diet 2, and 3 are acceptable; pls correct.

            -Crude protein Re-check %CP estimation of diets is an outage (32.55, 27.66% for diets) of the common range (CP 12-18%), and diet 1 is too low according to beef cattle requirement NRC or Brazil recommendation.

            -RDP re-check RDP estimation because the data is out of the commendation eg. NRC or Brazilian Tables of Food Composition for Ruminants

            -NPN re-check NPN estimation because the data is an outage the commendation eg. NRC or Brazilian Tables of Food Composition for Ruminants

            -ADF re-check ADF estimation because the data is an outage the commendation eg. NRC or Brazilian Tables of Food Composition for Ruminants

We improved and corrected Table 2. Please, see the revised version (L140). Some values are different from the nutritional recommendation. However, such information represents the diets used by this specific farm.

L170 adds the assessment impact from purchased feed categories because feeding contributes a large portion.
The impact from feed categories assessed purchased feed and the feed produced (grown) on farm. 

L182-186 describe how you consider CO2 assessed from manure, purchased feed, fertilizer, electricity, and fuels.
Manure: according to chapter 10 of IPCC (2006) (we provided a supplementary Table, please check). Purchased feed, fertilizer, electricity and fuels came from EcoInvent database, in the SimaPro software. It’s possible to consult and use lots of LCA from different products when evaluating a diet by its chemical composition

L220 Table 3

1)Suggest separating result Table eg. Carbon footprint, acidification, eutrophication with differing sources impact category

2)Carbon footprint presents data that categorized GHG sources (e.g. enteric methane fermentation, manure, purchased feed, fertilizer, electricity, fuels

3) Suggest to the discussion that the values are relatively low compared with other

We improved Table 3 as recommended, please see the revised version.  

L235-239 provide the result in the Table 2
Added.

L2243 Table 5 cannot be read, pls correct the presentation.
Table 5 was improved as suggested. 

L245 -249 move to GHG emission result section
Perfect, corrected.

L250-253 Table 5 cannot be read, pls correct the presentation.
Reformulated and corrected.

L297-298 re-check Table 3 CFP estimation to include all main category source impact e.g.  (e.g. enteric methane fermentation, manure, purchased feed, fertilizer, electricity, and fuels.
We double-checked this information. The CFP includes all impacts that can be expressed in CO2 eq. Thank you.

L346 enteric fermentation change to GHG emission
corrected. 

L361-363 data do not show, pls present Table data.
Please, see the revised manuscript. We added this information.

We sincerely appreciate the valuable comments and suggestions from the Reviewer.  

  Best regards.

Round 2

Reviewer 1 Report

The maniscript was improved. Some specific comments:

Lines 65-66: attention to the formatting of "CH4" and the formulas of the other GHGs. Check throughout the manuscript.

Table 2. Gross energy of diet 2 and diet 3 seem to be a bit low, expressed as MJ / kg dry matter. Please check. Attention also to the unit of Gross Energy (delete "/" between "kg" and "DM".

Line 144 please add that the period considered was from the arrival of the animals to their sale to the slaughterhouse

Line 157. "2.269%" can be change as "2.27%"

line 266. give a name to ($/animal)

Reviewer 2 Report

I have gone through the article, and it has improved. The article now seems good and is now ready for publication.